# Towards Small Object Editing: A Benchmark Dataset and A Training-Free Approach

### Qihe Pan
Zhejiang University of Technology
Hangzhou, Zhejiang, China
panqihe996@gmail.com

### Zhen Zhao
University of Sydney
Sydney, Australia
zhen.zhao@sydney.edu.au

### Zicheng Wang
The University of Hong Kong
Hong Kong, China
xiaoyao3302@outlook.com

### Sifan Long
Jilin University
Changchun, Jilin, China
longsf22@mails.jlu.edu.cn

### Yiming Wu
The University of Hong Kong
Hong Kong, China
yimingwu0@gmail.com

### Wei Ji
National University of Singapore
Singapore
jiwei@nus.edu.sg

### Haoran Liang
Zhejiang University of Technology
Hangzhou, Zhejiang, China
haoran@zjut.edu.cn

### Ronghua Liang*
Zhejiang University of Technology
Hangzhou, Zhejiang, China
rhliang@zjut.edu.cn

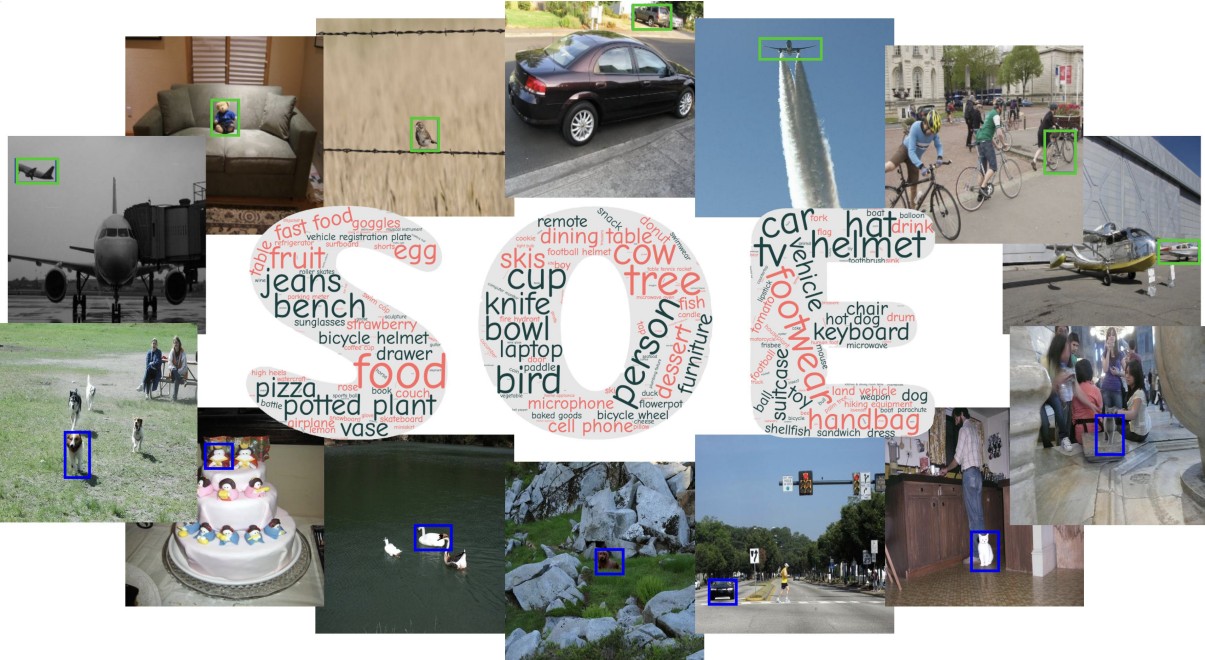

**Figure 1: We introduce SOEBench, a standardized benchmark for quantitatively evaluating text-based small object editing.**

*Corresponding author.

## ABSTRACT

A plethora of text-guided image editing methods has recently been developed by leveraging the impressive capabilities of large-scale diffusion-based generative models especially Stable Diffusion. Despite the success of diffusion models in producing high-quality images, their application to small object generation has been limited due to difficulties in aligning cross-modal attention maps between text and these objects. Our approach offers a training-free method that significantly mitigates this alignment issue with local and global attention guidance , enhancing the model's ability to

accurately render small objects in accordance with textual descriptions. We detail the methodology in our approach, emphasizing its divergence from traditional generation techniques and highlighting its advantages. What's more important is that we also provide *SOEBench* (Small Object Editing), a standardized benchmark for quantitatively evaluating text-based small object generation collected from *MSCOCO*[22] and *OpenImage*[18]. Preliminary results demonstrate the effectiveness of our method, showing marked improvements in the fidelity and accuracy of small object generation compared to existing models. This advancement not only contributes to the field of AI and computer vision but also opens up new possibilities for applications in various industries where precise image generation is critical. We will release our dataset on our project page: https://soebench.github.io/

## CCS CONCEPTS

• **Computing methodologies → Artificial intelligence**.

## KEYWORDS

Small Object Editing, Benchmark, Cross-Attention Guidance

**ACM Reference Format:**
Qihe Pan, Zhen Zhao, Zicheng Wang, Sifan Long, Yiming Wu, Wei Ji, Haoran Liang, and Ronghua Liang. 2024. Towards Small Object Editing: A Benchmark Dataset and A Training-Free Approach. In *Proceedings of the 32nd ACM International Conference on Multimedia (MM '24), October 28-November 1, 2024, Melbourne, VIC, AustraliaProceedings of the 32nd ACM International Conference on Multimedia (MM'24), October 28-November 1, 2024, Melbourne, Australia.* ACM, New York, NY, USA, 9 pages. https://doi.org/10.1145/3664647.3680896

## 1 INTRODUCTION

The realm of text-to-image generation has witnessed tremendous advancements with the advent of recent diffusion models[12, 25, 27, 29, 34], which have successfully revolutionized various tasks, including photo editing[15, 36], and inpainting[2, 30]. These models have demonstrated remarkable capabilities in producing and manipulating salient objects within a picture, like the main subject of a picture, under the description guidance. The success of such models can be attributed to the effective cross-modal feature alignment between textual descriptions and the corresponding visual objects during the synthesis process. Such alignment facilitates a coherent and accurate translation of textual descriptions into visual representations, resulting in impressive outcomes in both image editing and in-painting tasks.

However, a severe issue arises when the pending object is small. As shown in Fig. 2 (b), using small masks to guide the target object editing can lead to problems such as attribute leakage, poor quality, and missing entities. This issue is primarily due to the model's inability to focus on such a small region of interest in the description, which can result in the model's inability to generate objects that align well with the textual descriptions. For example, consider an image with a size of $512 \times 512$ and a small object whose bounding box typically occupies only $64 \times 64$ pixels. When performing multi-level cross-modal feature alignment, with U-Net as the backbone, the cross-attention map is progressively down-sampled to a small resolution of $8 \times 8$ as the network deepens. At the same time, our

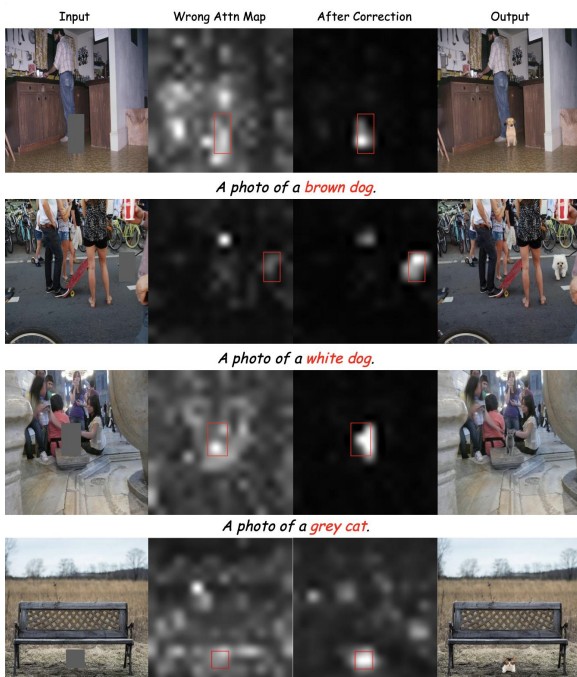

**Figure 2: Comparison between our multi-scale joint attention guidance method and traditional text-based image in-painting methods. Traditional text-based image in-painting methods can only edit objects with a large scale. The first column: The input image with mask condition. The second column: Small mask condition is more likely leads to wrong attention map. The third column: Our method multi-scale joint attention guidance method can obtain the refined cross-attention map for accurate small object editing. The last column: The output image generated by refined cross-attention map.**

target region may encompass within only a $1 \times 1$ grid, which is too small for the model to effectively focus on. Consequently, the model may struggle to generate objects that align well with the textual descriptions within such a small area, leading to poor quality and even missing entities.

Facing the abovementioned limitations, we introduce a new task called Small Object Editing (SOE) in this work. Specifically, the SOE task requires the model to perform editing that is consistent with the given textual descriptions, seamlessly integrates with the surrounding context of the original image, and is precise in the desired small region. The key to this task is to prevent mismatches between the desired masked region and the intended textual description, ensuring that the model generates accurate and high-quality small objects. To evaluate the performance of models on this SOE task, we construct a comprehensive benchmark dataset, *i.e.*, SOEBench. This benchmark allows us to evaluate the effectiveness of small object editing on various models and includes 4000 objects from two established datasets, MSCOCO [22] and OpenImages [18]. SOEBench holds two sets, SOE-2k and SOG-4k, where SOE-2k contains 2000

objects from OpenImages for editing and SOE-4k contains an additional 2000 objects from MSCOCO for editing. For each small object editing prompt, we provide both label-only and label-with-color templates in the description. This allows us to evaluate the models' ability to generate small objects that align with both textual descriptions and color specifications.

Building upon the constructed SOEBench, we further provide a strong baseline method for small object editing. As discussed above, the quality of the cross-attention map is crucial in small object editing. To this end, we propose a new joint attention guidance method to enhance the accuracy of the cross-attention map alignment from both local and global perspectives. In particular, we first develop a local attention guidance strategy to enhance the foreground cross-attention map alignment and then introduce a global attention guidance strategy to enhance the background cross-attention map alignment. Our proposed baseline method is training-free but highly effective in addressing the SOE problem.

The contributions of our work are summarized as below:

- By recognizing the limitations of current research, we define a new task of small object editing and come up with a new benchmark dataset SOEBench to evaluate the model's ability on small object editing.
- We introduce a novel training-free baseline approach to address the small object editing challenge by considering both global and local perspectives.
- The proposed method is evaluated on the proposed SOEBench dataset and achieves the state-of-the-art results.

## 2 RELATED WORK

### 2.1 Diffusion Models

Denoising diffusion based models (DMs)[12, 25, 27, 29, 31] and score-matching-based model[32–34] have become the *de facto* standard for image generation, surpassing Generative Adversarial Networks (GANs)[9, 35, 38, 39, 41] in performance and training stability. Denoising Diffusion Probabilistic Models (DDPM)[12] and Noise Conditional Score Networks (NCSN)[34] are the pioneer to employ denoising neural networks to reverse a predefined Markovian noising process on raw images.

Latent Diffusion Models (LDM)[29] intuitively disentangle image synthesis into two stages: semantic reconstruction through a denoising process in latent space, and perceptual reconstruction with VAE[17], such a strategy significantly enhances visual fidelity. Currently, the success of DMs has greatly driven a series of image generation and editing tasks.

### 2.2 Text-based Image Editing

Image editing has become a popular domain, enabling personalized and customized image generation. Text prompts are predominantly employed as input conditions due to their flexibility. These methods can generally be classified into two categories: training-based and training-free. Training-based methods typically involve fine-tuning the entire denoising model or adapters. For example, Diffusion-CLIP [16] and InstructPix2Pix [4] fine-tune the entire diffusion model using collected text-image datasets. In contrast, Asyrp [19]

and GLIGEN [21] introduce adapters into publicly available diffusion models and only tune the adapter parameters for rapid adaptation. T2I-Adapter [24] and ControlNet [40] further extend control capabilities by introducing tailored adapter architectures such as zero convolution.

For training-free methods, manipulating cross-attention maps and fine-tuning text embeddings are mainstream techniques. Prompt-to-prompt [10] edits images solely through text prompts, injecting attention maps of the original image throughout the diffusion process. Attend-and-Excite [5] introduces GSN to intervene in the diffusion process to enhance subject tokens for generating multiple objects. DDIM Inversion [31] presents a deterministic process to invert images to noises. Inspired by DDIM Inversion, Null-Text Inversion [23] pioneers fine-tuning null-text embeddings to reduce the distance between the sampling trajectory and the inversion trajectory. Prompt Tuning Inversion [8] follows a two-stage pipeline, first encoding the image into a learnable embedding and then sampling the edited image by interpolating the target embedding and the learnable embedding. PnP Inversion [14] enhances the inversion process by disentangling source and target branches for content preservation and edit fidelity, respectively. Imagic [15] incorporates text embedding optimization and model fine-tuning for image editing. BlendedDiffusion [1, 2] explore the method of using a mask to edit the specific region and add a new object to the image while leaving the rest unchanged. Some layout2image methods[6, 7, 26, 37] operated constraints on cross-attention to control the synthetic contents.

Most text-based image editing methods are based on DMs, while a U-Net architecture is widely adopted. However, as the U-Net deepens, the cross-attention map becomes extremely small, thus some small objects may only take a small region on the attention map, leading to the model hardly focusing on such small objects for editing.

### 2.3 Benchmarks for Image Editing

Although image editing has been widely explored in recent years[3, 13–15, 36], benchmarks for this task remain limited. EditBench [36] primarily presents a systematic benchmark comprising 240 images for image inpainting based on masks and text prompts, evaluated using CLIP-Score[11, 28] and CLIP-R-Precision metrics. Concurrently, Kawar *et al.* introduce TedBench [15] to evaluate non-rigid text-based image editing, providing 100 pairs of input images and target texts, with CLIP-Score and 1-LPIPS adopted as metrics. Edit-Val [3] introduces a standardized evaluation protocol for assessing multiple edit types using a curated image dataset. Additionally, Ju *et al.* present PIE-Bench [14], comprising 700 images showcasing diverse scenarios and editing types. Aside from automatic evaluation via benchmarks, human evaluation (*a.k.a.* user study) is a more reliable method for assessing the quality of generated images. *Text Alignment* and *Image Quality* are two main metrics used in these studies. *Text Alignment* focuses on the consistency between the text prompt and the edited image, while *Image Quality* evaluates the visual fidelity of the generated image.

However, to the best of our knowledge, there is no benchmark so far focusing on small object editing, which is a critical issue failed by most of the current research. Therefore, in this work, we focus

on small object editing and propose a new small object editing benchmark, *i.e.*, SOEBench.

## 3 PRELIMINARIES

### 3.1 Latent Diffusion Models

Latent diffusion models are a subclass of generative models that adopt a diffusion and a denoising process to synthesize new data. First, a VAE encoder is adopted to encode the image $\mathcal{I}$ into the latent space, denoted as $z$. Then the forward diffusion process gradually introduces noise to the latent representation $z$ to the complete Gaussian noise $z_T$, where $T$ represents the total number of timesteps. For each timestep $t$, the noisy latent code $z_t$ can be represented as:

$$z_t \sim \mathcal{N}(\alpha(t)z; \sigma^2(t)\mathbf{I}), \tag{1}$$

where $\alpha(t)$ and $\sigma(t)$ control the mean and covariance of the noise. Then, the diffusion models aim to reverse this diffusion process by sampling random Gaussian noise $z_T$ and gradually denoising such latent code to generate the initial latent code $z_0$. In practice, the denoising model targets at predicting the sampled noise $\epsilon_\theta$ at each timestep $t$ given the condition $c$ by optimizing the Mean Square Error between the predicted sampled noise $\epsilon_\theta(z_t, c, t)$ and the sampled noise $\epsilon$, which can be formulated as:

$$\mathcal{L} = \nabla_\theta \|\epsilon - \epsilon_\theta(z_t, c, t)\|^2. \tag{2}$$

Finally, a VAE decoder is adopted to generate the image using the denoised latent code.

### 3.2 Cross-Attention Guidance

The Latent Stable Diffusion models perform conditional generation utilizing a cross-modal attention module between the given conditioning $c$ and the latent representations $z_t$, where the condition embedding $c$ and the latent representations $z_t$ are mapped to the query $\mathbf{Q}$ with a dimension of $M \times d$ and the key $\mathbf{K}$ with a dimension of $N \times d$, and the cross-attention map $\mathcal{A}^l$ in the $l$-th layer of the U-Net can be derived as:

$$\mathcal{A}^l = Softmax\left(\frac{\mathbf{QK}^T}{\sqrt{d}}\right) \in [0, 1]^{M \times N}. \tag{3}$$

Attention control is widely used in image editing, notably in techniques such as Prompt-to-Prompt(P2P). P2P replaces the current attention map with one corresponding to the target text, thereby facilitating the editing of objects within the image.

## 4 THE SOEBENCH DATASET

SOEBench is selectively extracted from MSCOCO[22] and Open-Image[18] databases, the selection rule of which follows a set of precise criteria to aptly serve the needs of small object generation experiments. The selection process primarily focuses on ensuring that the chosen objects are not obstructed by other elements within the image, allowing for a clear target for the small object generation task. Additionally, the object size was a critical consideration. We specifically selected objects that occupy a size smaller than 1/6 but larger than 1/8 of the overall image. This size constraint is pivotal because objects smaller than 1/8 would yield a representation on the deepest U-Net's feature map that is smaller than one pixel, rendering them practically unfeasible for effective generation, as shown in Fig. 4.

The category of the data contained in the dataset can be referred to in the word cloud in Fig 1. Our dataset comprises about 300 types of objects that are frequently encountered, aligning the experimental setup with real-world object recognition scenarios. We then crop the images to extract the portions within the masked areas, and then use the BLIP-VQA [20] model to query, 'What is the primary color of the object in this area?' This approach efficiently identifies the main color of the objects within the masks as the color attribute. Importantly, the dataset has been segmented into two subsets based on the quantity of images they contain: *SOE-2k* and *SOE-4k*, where *SOE-2k* contains 2000 objects from the OpenImage dataset and *SOE-4k* contains 2000 more objects from the MSCOCO dataset. Such an operation enables a thorough and diverse assessment, ensuring that our methodology is tested under different scales of data availability, further reinforcing the validity and robustness of our research findings.

**Table 1: Comparing our benchmark with existed works from different aspects, such as object category, benchmark size and mask size.**

| Benchmarks | Benchmark Info | | |
|---|---|---|---|
| | categoty num | dataset size | mask size |
| TedBench | <50 | 100 | >0.5 |
| EditBench | <50 | 240 | 0.08~0.9 |
| EditVal | <50 | 648 | - |
| SOE(Ours) | 300 | 2K/4K | <0.03 |

Compared to existing benchmarks for text-based image editing (*i.e.,* TeDBench [15], EditBench [36], and EditVal[3]), our benchmark, SOEBench, offers distinct advantages. TeDBench is limited by its small dataset, consisting of only 100 images across 40 categories, primarily focused on modifying object attributes, states, or appearances. Furthermore, TeDBench images typically feature a single dominant subject occupying a significant portion of the image. In contrast, EditBench includes a larger dataset of 240 images and accommodates a wider range of mask sizes, from 0.08 to 0.9 of the image size. However, even EditBench's smallest mask sizes are substantially larger than those in SOEBench, where the maximum size of small object areas occupies only $\frac{1}{36}$ of the image. Another recent benchmark, EditVal[3], offers a more comprehensive set of image-editing operations. Developed from the MSCOCO [22] dataset, EditVal comprises 648 unique image-editing operations across 19 classes. These operations include 13 types of real-world edits, including one dedicated to controlling object size. Comparing SOEBench with these benchmarks (summarized in Tab. 1), we observe that SOEBench offers a larger and more diverse test dataset, covering a wider array of categories. Moreover, SOEBench features significantly smaller target areas, presenting a greater challenge for text-based image editing tasks.

## 5 METHODOLOGY

In this section, we present our training-free multi-scale joint attention guidance baseline in detail. In particular, we first describe the problem definition of small objection editing in Sec. 5.1. Then, we

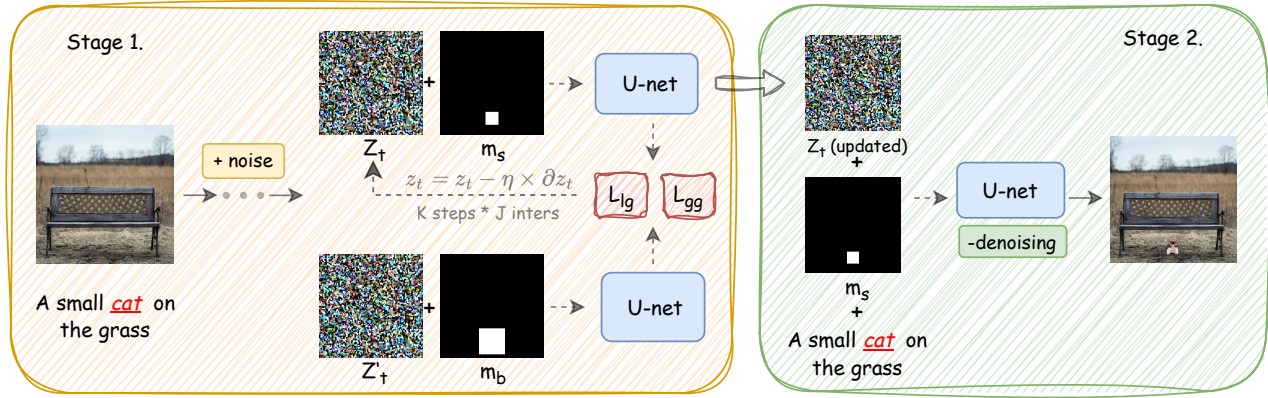

**Figure 3: The inference pipeline of our proposed *SOE*. We receive smaller mask region $m_s$, larger mask region $m_b$ then randomly initialize two identical $z_T$ and $z'_T$, text prompt $c_t$ as input. During the first K timesteps, we compute the cross-attention maps from both parts J times at each timestep and calculate the $\mathcal{L}_{lg}$ and $\mathcal{L}_{gg}$ losses. Then, based on the loss gradients, we backpropagate to optimize $z_t$. After $K \times J$ rounds of optimization, we continue to use the diffusion model to denoise the image.**

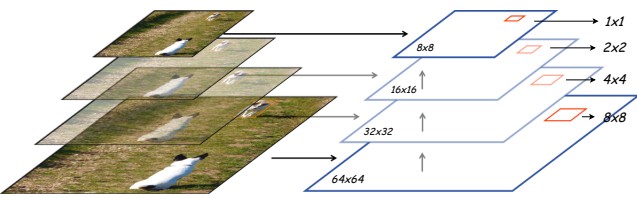

**Figure 4: The illustration of a cross-attention map. For an input image with a size of $512 \times 512$, if the masked area is $64 \times 64$, the corresponding effective area comes to $1 \times 1$ in the mid-block. The diminutive size of this masked area poses a challenge as it may lack sufficient semantic information essential for generating associated objects.**

introduce our local attention guidance in Sec. 5.2 and our global attention guidance in Sec. 5.3.

## 5.1 Problem Definition

In this paper, we propose a novel small object editing (SOE) task that targets at in-painting the specified masked region $m_s$ on the given image $I$, which is usually small, using a textual description $c$ as the condition. The task requires the model $\Psi$, which includes a VAE encoder $\mathcal{E}$, a diffusion model with weight $\theta$ from which we can obtain the predicted noise $\epsilon$ and cross-attention map $\mathcal{A}$, and a VAE decoder $\mathcal{D}$, to not only perform inpainting consistent with the given condition but also perform the in-painting that integrates seamlessly with the surrounding context of the original image. Following previous works, a $L$-layer U-Net is adopted as the backbone of the diffusion model. The key to the task is to avoid any mismatch between the desired masked region and the intended textual description, which is usually achieved by refining the cross-attention map $\mathcal{A}$.

In this paper, we aim at proposing a training-free baseline method to tackle the SOE problem. Based on the $L$-layer U-Net architecture, we focus on cross-attention map refinement and propose a local and global attention guidance method, where we encourage the refined

cross-attention map to provide precise guidance to the latent code estimation. The overall inference pipeline is shown in Fig. 3.

## 5.2 Local Attention Guidance

During the inference stage, given the small mask $m_s$ and the image $I$, we implement local attention guidance according to the following steps. **1)** we scale the height and the width of $m_s$ according to the scaling factor $s$ to obtain a large mask $m_b$. Note the center position of $m_b$ is identical to the center position of $m_s$. **2)** Two sets of attention map $\mathcal{A}^l$ and $\mathcal{A}'^l$ can be obtained according to Eq. 4 and Eq. 5, where

$$\mathcal{A}^l \leftarrow \mathcal{F}_\theta(z_t, t, c, m_s, z), \tag{4}$$

and

$$\mathcal{A}'^l \leftarrow \mathcal{F}_\theta(z'_t, t, c, m_b, z), \tag{5}$$

where $\mathcal{F}_\theta$ is the cross-attention map obtainment operation, where we collect the cross-attention maps from the diffusion model, $z'_t$ is the noised latent code predicted by using $m_b$ as the input, which is initialized by $z'_T = z_T$ and $l$ denotes the layer index of the model. **3)** We use the index of the label $c_{req}$ in the given text condition $c$, *e.g.*, we obtain the index of the label "cat" in the given text "A small cat on the grass", to index the attention maps desired, *i.e.*, $\mathcal{A}^l_{c_{req}}$ and $\mathcal{A}'^l_{c_{req}}$. **4)** We crop $\mathcal{A}^l_{c_{req}}$ and $\mathcal{A}'^l_{c_{req}}$ according to $m_s$ and $m_b$, respectively, to obtain the desired attention maps $r_{m_s}(\mathcal{A}^l_{c_{req}})$ and $r_{m_b}(\mathcal{A}'^l_{c_{req}})$. **5)** We rescale $r_{m_b}(\mathcal{A}'^l_{c_{req}})$ using bilinear interpolation to the same size as $r_{m_s}(\mathcal{A}^l_{c_{req}})$ to calculate $L_{lg}$ loss according to Eq. 6.

$$\mathcal{L}_{lg} = \sum_{l=1}^{L} \sum_{i \in \{c_{req}\}} \left\| I(r_{m_b}(\mathcal{A}'^l_i)) - r_{m_s}(\mathcal{A}^l_i) \right\|_2, \tag{6}$$

Based on our observation, the model is prone to generate accurate contents within such a large region. However, such an operation may fail to handle the seamless integration with the surrounding context of the original image problem. Therefore, we just

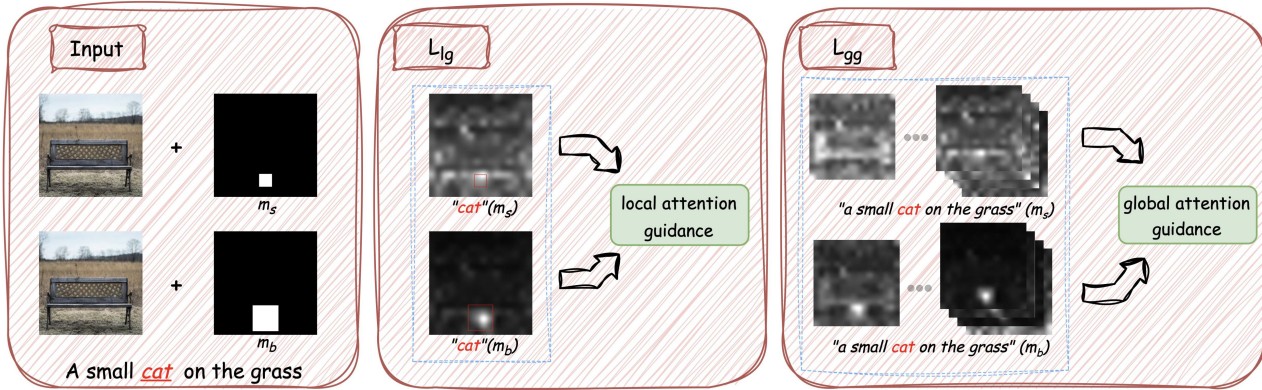

**Figure 5: Local attention is primarily used to correct the semantic information within the mask area of the $\mathcal{A}$ corresponding to the main object, ensuring it aligns with what is correct. Global attention guidance collect $\mathcal{A}$ from all work tokens and primarily aims to accurately distinguish unimportant details beyond the main object during the process of generating the $\mathcal{A}$ from the latent code.**

rescale the $\mathcal{A}'^l$ and encourage such a precise cross-attention map to provide guidance for $\mathcal{A}^l$.

It should be mentioned that Eq. 6 is designed solely to modify $\mathcal{A}^l$ while $\mathcal{A}'^l$ remains unchanged, thus obviating the need for a back-propagation operation.

## 5.3 Global Attention Guidance

Our local cross-attention guidance approach is training-free but boosts the accuracy of the desired region in the cross-attention map, which shows its great potential in tackling the SOE problem. However, as shown in Fig. 5, as the desired mask region may be too small, the pixels on the cross-attention map may be attracted to the wrong region, like the background. To tackle the issue, in this section, we further propose a global attention guidance operation to alleviate the foreground noise, which guarantees that when utilizing $m_s$ to produce a cross-attention map, the map accurately zeroes in on pertinent information within the masked areas associated with the object being generated, effectively discounting data from areas inconsequential to the textual depiction of the object.

In particular, based on $\mathcal{A}'^l$ introduced in Sec.5.2, we encourage $\mathcal{A}'^l$ to provide global guidance for $\mathcal{A}^l$ to alleviate the problem of cross-attention matching to the wrong region. Different from local attention guidance that only focuses on the cross-attention map on the desired mask region alignment, we encourage $\mathcal{A}^l$ to be aligned with $\mathcal{A}'^l$ for all queries, which can be formulated as

$$\mathcal{L}_{gg} = \sum_{l=1}^{L} \sum_{i=1}^{I} \left\| \mathcal{A}'^l_i - \mathcal{A}^l_i \right\|_2 . \tag{7}$$

Recall that $i$ is the index for the $i$-th condition token, $I$ represents the length of the sentence. Also note that $\mathcal{A}'^l$ is still unchanged and only the value of $\mathcal{A}^l$ needs to be modified, and no back-propagation operation is required.

The overall guidance function is defined as:

$$\mathcal{L}_{total} = \mathcal{L}_{lg} + \mathcal{L}_{gg}. \tag{8}$$

Then, we compute the gradient of $z_t$ and update the predicted $z_t$ as:

$$z_t \leftarrow z_t - \eta \nabla_{z_t} \mathcal{L}_{total}, \tag{9}$$

where $\eta$ is the learning rate.

Based on our experiments, we found that we only need to perform our joint attention guidance on the cross-attention map for the former $K$ denoising timesteps, where $z_t$ undergoes a series of updates. Specifically, it is refined through a gradient descent process aimed at minimizing $\mathcal{L}_{total}$, which occurs $J$ times. The step-by-step methodology for our method is delineated in Algorithm 1.

Our joint attention guidance method performs cross-attention map alignment from a local perspective and a global perspective, which provides a comprehensive correction mechanism. This method aims to enhance the model's ability to accurately focus and generate small targets that are consistent with text descriptions, and assign lower attention values to non-target areas, thereby addressing the key challenge of using diffusion models to generate small targets.

## 6 EXPERIMENTS

### 6.1 Evaluation Metrics.

In the evaluation of our methodology, we employed two metrics: the CLIP-Score and the Fréchet Inception Distance (FID). The CLIP-Score was used to quantify the similarity between the generated objects within the target area and the corresponding text descriptions. This metric effectively assesses how well our model's output aligns with the textual prompts, particularly focusing on the small objects generated. The FID Score was utilized to gauge the quality of the generated images in relation to the original images. Higher CLIP-Score or lower FID, indicate better performance.

It's important to note that the generated objects occupy a relatively small proportion of the original image, using the entire image for metric calculation could lead to misleading results. Therefore, we apply a targeted approach by cropping the original images during the evaluation phase. This cropping is designed to focus on the

---

**Algorithm 1:** Our joint attention guidance method

---

1 **Input:** the textual prompt $c$, the region mask $m_s$, the extracted latent code $z$, the number of total timestep $T$, the number of guiding timestep $K$, the number of backward times $J$, the diffusion model $\epsilon_\theta$, the cross-attention map obtainment operation $\mathcal{F}_\theta$, the learning rate $\eta$.

2 **Output:** the estimated latent code $z_0$.

---

3 **Initialization:** $z_T \sim \mathcal{N}(0, I)$, $z'_T \leftarrow z_T$

4 **Scale:** $m_b \leftarrow$ enlarge $m_s$. # Fix the center of the small box, then enlarge the length and width by a factor of $s$.

5 **for** $t = T, \ldots, T - K$ **do**

6      $z'_{t-1} = \epsilon_\theta(z'_t, t, c, m_b, z)$ , $\mathcal{A}'^l = \mathcal{F}_\theta(z'_t, t, c, m_b, z)$

7      **for** $j = 0, \ldots, J - 1$ **do**

8          $z_{t-1} = \epsilon_\theta(z_t, t, c, m_s, z)$ , $\mathcal{A}^l = \mathcal{F}_\theta(z_t, t, c, m_s, z)$

9          **Get Features to Compute the** $L_{lg}$**:**

10          $\mathcal{A}^l_{c_{req}} = \mathcal{A}^l[c_{req}]$ , $\mathcal{A}'^l_{c_{req}} = \mathcal{A}'^l[c_{req}]$

11          $r_{m_s}(\mathcal{A}^l_{c_{req}})$, $r_{m_b}(\mathcal{A}^l_{c_{req}})$ #features within $m_s$, $m_b$

12          **Rescale:**

13          $I(r_{m_b}(\mathcal{A}'^l_{c_{req}})) \leftarrow$ resize $r_{m_b}(\mathcal{A}'^l_{c_{req}})$ downscaling the $r_{m_b}(\mathcal{A}'^l_{c_{req}})$ using bilinear interpolation .

14          $\mathcal{L}_{total} = \mathcal{L}_{lg}(r_{m_s}(\mathcal{A}^l_{c_{req}}), I(r_{m_b}(\mathcal{A}'^l_{c_{req}}))) + \mathcal{L}_{gg}(\mathcal{A}^l, \mathcal{A}'^l)$ , $\nabla_{z_t} = \partial_{z_t} \mathcal{L}_{total}$ , $z_t = z_t - \eta \nabla_{z_t}$

15      **end**

16 **end**

17 **for** $t = T - K - 1, \ldots, 0$ **do**

18      $z_t = \epsilon_\theta(z_t, t, c, m_s, z)$

19 **end**

20 **Return:** $z_0$

---

area of interest, ensuring a more direct and meaningful comparison between the generated object and the original context. This method of evaluation ensures that our metrics accurately reflect the model's performance in generating small objects, providing a true assessment of its capability in this specific task.

## 6.2 Implementation Details

We use the stable diffusion v1-5 painting model as the baseline method and use DDIM as the noise scheduler. Considering the resize operation on the mask area, a too-large resize ratio will bring a large loss error. We set the scaling factor $s$ to a value randomly selected from 1.5 to 3. We set the learning rate $\eta = 100$ by default. The noise correction is performed during the initial $K = 5$ steps of the denoising process and repeated $J = 5$ times at each step. More ablation study about the hyper-parameter can be found in Tab.XX, and the results indicates that our method is not sensitive to the hyper-parameters, highlighting the robustness of our approach.

## 6.3 Experimental Results

We evaluate our approach on our proposed benchmarks *SOE-2k* and *SOE-4k* with two kinds of prompt templates, *i.e.*, label and

**Table 2: Compare our proposed training-free approach with the stable diffusion model (SD-I) on the SOEBench dataset. lg indicates our local attention guidance method and gg indicates our global attention guidance method. The best results are highlighted in bold.**

| Prompt | Methods | SOE-2k | | SOE-4k | |
|---|---|---|---|---|---|
| | | CLIP-Score | FID | CLIP-Score | FID |
| label | SD-I | 23.79 | 34.65 | 23.11 | 35.21 |
| | SD-I+lg | 23.93 | 35.02 | 23.25 | 35.02 |
| | SD-I+lg+gg | **24.05** | **34.32** | **23.39** | **34.67** |
| color +label | SD-I | 24.10 | 34.73 | 23.31 | 34.96 |
| | SD-I+lg | 24.21 | 34.32 | 23.47 | 34.74 |
| | SD-I+lg+gg | **24.36** | **33.85** | **23.62** | **34.28** |

color+label, *e.g.*, "a dog" and "a brown dog", and quantitatively compute the CLIP-Score and FID to prove the effectiveness of our cross attention correction. The result is shown in Tab. 2 and our method demonstrated a great improvement compared to the baseline models, exceeding the baseline method stable diffusion by a large margin of nearly 1 point in terms of the FID score on the *SOE-2k* subset, and around 0.7 points in terms of the FID score on the *SOE-4k* dataset. It can be also inferred from the table that given a detailed condition, *i.e.*, color with label, the editing performance of our method exceeds the performance of given a simpler condition, *i.e.*, label only, by a large region of around 0.5 points in terms of the FID score on the *SOE-2k* subset, indicating the effectiveness of our method of understanding different conditions. It should also be mentioned that our method is training-free, and the experimental results verify the great potential of our method on tackling the small object editing problem.

## 6.4 Ablation Study

We further analyze the effectiveness of the detailed module designs of our joint attention guidance method, which includes a local attention guidance (lg) method and a global attention guidance (gg) method. It can be inferred from Tab. 2 that our local attention guidance can efficiently improve the models' in-painting performance on the small object editing task, mainly attributed to our newly proposed mask region scaling-and-rescaling operation, which efficiently enables the model to generate precise content corresponding to the textual description, and can also be integrated with the surrounding context of the original image successfully. Based on our local attention guidance method, our global attention guidance method can further bring performance improvement, as our global attention guidance method further reduces the probability of the model modifying the background part, ensuring the accuracy of the modified part, and at the same time ensuring the quality of background image generation.

## 6.5 Qualitative results

To complement our quantitative studies above, we present qualitative results in this section. We provide samples and qualitative comparison with Stable-Diffusion Inpainting model in Fig 6. As show in the examples, when using the SD-I model for generating images of small objects, issues frequently arise such as low quality, mismatches between text and images, or the absence of generated objects. However, as shown in the third column of each image group

in the figures, our method has been able to address these issues to a considerable extent, enhancing the performance of the base model.

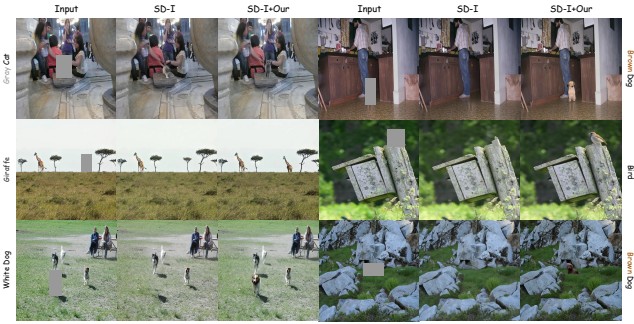

**Figure 6: Qualitative results of our method on the SOEBench dataset. The first column of each set contains the original image, mask, and text. The second column shows results from the SD-I model, and the third column displays outcomes of SD-I+Ours.**

## 6.6 User Preference Study

Besides the automated metrics, we also incorporated a user study to align more closely with intuitive human preference. In the study, we aim to assess both prompt adherence and the overall image. We employed our methods and SD-I(Stable-diffusion-Inpainting) as comparison models, keeping the seed fixed to generate 25 pairs, totaling 50 images. Our study involved 100 total participants. Participants were tasked with ranking images of small target objects generated these two methods. Fig. 7 present the user study. The most important results are:Our training-free methods can enhances the base model's ability to improve both the image quality of generated objects and their alignment with the associated text to a certain extent.

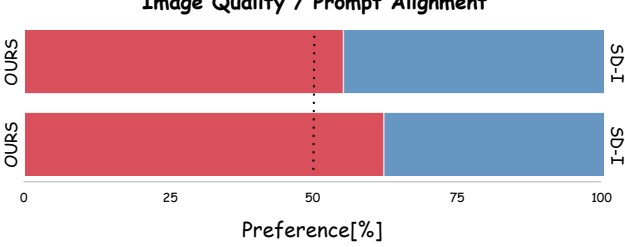

**Figure 7: User preference study. We compare the performance of our training-free methods against baselines Stable-Diffuion Inpainting. Our method outperforms baseline in both image quality and prompt alignment study.**

## 6.7 Discussion

**Method comparison with P2P.** In the realm of image content editing, prompt-to-prompt (P2P) methods have proven to be highly effective for various image manipulation tasks. Nevertheless, they exhibit notable limitations in generating small objects, primarily due to the mismatch between text and the corresponding cross-attention map. To further evaluate this, we conducted tests using

a P2P-based method, wherein we attempted to transfer the appropriate attention map from the large mask to the smaller one. Unfortunately, this approach proved ineffective for the generation of small objects because it did not correct the erroneous regions of the cross-attention map. This further corroborates the effectiveness of our method in addressing this specific issue.

**Failure Cases.** While our method effectively mitigates performance degradation through attention guidance, its efficacy remains constrained by the limitations of the base model. As depicted in Fig. 8, we showcase instances of object omission and subpar quality in our generated images. To bolster the small object editing capabilities further, fine-tuning the model on dedicated small object images presents a straightforward solution.

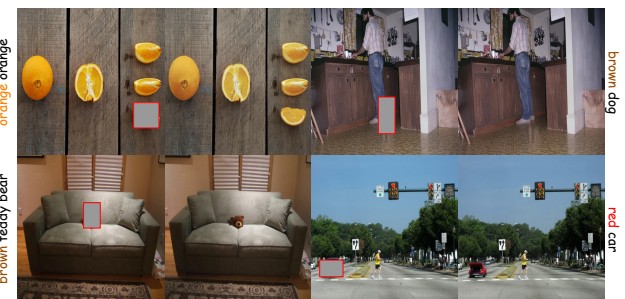

**Figure 8: Failure cases. Due to the performance of the base model, issues such as object missing and low quality will still be encountered even our joint attention guidance approach is adopted.**

## 7 CONCLUSION

In this work, our research introduces two significant contributions to the field of *SOE* (Small Object Editing) using diffusion models. We firstly developed a new benchmark dataset specifically designed to evaluate small object editing. This dataset addresses the unique challenges and requirements of small object imagery, providing a comprehensive and targeted platform for testing and comparing different models. Alongside this dataset, we propose a training-free approach represents a major advancement in solving the issue of generating small objects. By focusing on the alignment of cross attention maps between text and small objects, our method effectively bridges the gap that has historically hindered accurate and detailed generation of small objects in response to textual descriptions. This alignment enables the model to more precisely interpret and render the intricate details required for small object generation .This baseline serves as an initial reference point for future research, offering a simplified yet effective framework for subsequent models to build upon and refine.

Another critical aspect to consider is that our method operates within the constraints of the existing capabilities of the diffusion model's backbone. Since we do not modify the parameters of the diffusion backbone, the quality and the fidelity of the generated images are inherently bound by the performance of the underlying model. Therefore, while our approach enhances the model's ability to generate small objects in alignment with text descriptions, the overall effectiveness is still dependent on the inherent capabilities of the diffusion model.

## ACKNOWLEDGMENTS

This work is supported by the National Natural Science Foundation of China (No.62202431).

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
