# OpenReview forum: "Towards Small Object Editing: A Benchmark Dataset and A Training-Free Approach"
_acmmm.org/ACMMM/2024/Conference — MM2024 Poster_

### Official Review · Reviewer_k18F · 2024-05-15

**Rating:** 3
**Confidence:** 3

**Summary:**

This paper explores text-guided image editing methods, particularly focusing on small object generation, using large-scale diffusion-based generative models. The method offers a retraining-free solution to align cross-modal attention maps between text and small objects, thereby improving the accuracy of rendering objects according to textual descriptions. Additionally, this paper constructs the SOEBench dataset, which serves as a standardized benchmark for quantitatively evaluating text-based small object generation. The research problem addressed in the paper is highly practical, and the proposed method is closely tied to the research problem. However, the authors lack experimentation, especially in comparison with other methods.

**Strengths:**

1) The research problem is well-defined and has strong practical value. The proposed method effectively addresses the issue of small target editing in text-to-image tasks.
2) A new image editing method for small targets is proposed, enabling content editing in images without the need for retraining.
3) A Benchmark Dataset for small target editing is constructed, which surpasses existing datasets in terms of data scale and category richness.
4) Comprehensive analysis and discussion of the proposed method are provided, although comparison with existing methods is lacking.

**Limitations:**

1) Figure 1, serving as the abstract figure, only annotates some small target regions without conveying the research methodology and contributions of the paper.
2) Experimental results mainly focus on the proposed method without comparison with relevant methods. However, research on text-to-image editing is extensive, highlighting a significant drawback of this paper.
3) Section 6.7 discusses a comparison with P2P methods but relies solely on textual discourse, making it difficult to convince readers; incorporating visual experiments would better illustrate the method's advantages.
4) The reference format is inconsistent, for instance, using abbreviations for CVPR in [4] while providing the full name for the reference [2] (In Proceedings of the IEEE/CVF Conference on Computer Vision and Pattern Recognition), and similar issues are present in many references.
5) It is recommended to include discussions on time complexity in the method analysis in the experimental section for a more comprehensive discussion of method practicality.

**Suitability:**

3

---

### Official Review · Reviewer_UTA5 · 2024-05-26

**Rating:** 5
**Confidence:** 3

**Summary:**

This manuscripts offers a dataset for small object editing. In addition, a training-free approach is proposed to address small object editing.

**Strengths:**

The benchmark has rich amount of data whose mask size is specifically targeting the small size. The training-free approach to implement small object editing is simple and innovative.

**Limitations:**

1. It would be better to offer the full-resolution image or zoom-in version of the small object to evaluate the quality of the small object.
2. It would be better if the manuscript can add Adobe Photoshop's generative fill's results in user preference study.

**Suitability:**

3

---

### Official Review · Reviewer_CaZu · 2024-06-09

**Rating:** 4
**Confidence:** 2

**Summary:**

The manuscript presents a significant contribution to the field of text-guided image editing, particularly focusing on the challenging task of small object editing. The authors propose a novel, training-free approach that leverages the powerful capabilities of large-scale diffusion-based generative models, specifically Stable Diffusion. They introduce a comprehensive benchmark dataset, SOEBench, designed to evaluate the performance of text-based small object editing models quantitatively.

**Strengths:**

1. The proposed method enhances cross-modal attention alignment through local and global attention guidance, which significantly improves the model's ability to accurately render small objects based on textual descriptions.

2. The authors present extensive experimental results, including quantitative metrics and qualitative examples, to validate their approach. The improvement in performance over baseline models is clearly demonstrated.

3. The user preference study adds a practical dimension to the research, highlighting the real-world applicability and user satisfaction with the generated outputs.

**Limitations:**

1. The description of the local and global attention guidance methods lacks clarity in some areas. Specifically, in Section 5.2 (Local Attention Guidance), the process of "scaling-and-rescaling" the target region is not thoroughly explained. It would be beneficial to include more detailed steps or a pseudo-code to help readers understand how the mask regions are upscaled and integrated into the cross-attention map.

2. The paper does not provide sufficient information on the choice of hyperparameters such as the scaling factors, the learning rate η, and the number of timesteps K and J in Algorithm 1. Including a discussion on how these hyperparameters were selected and their impact on the results would enhance reproducibility.

3. The method's reliance on the accuracy of the initial cross-attention map may limit its effectiveness if the initial attention map is significantly misaligned. The authors should discuss how robust their approach is to variations in the initial map quality.

4. The user preference study, while valuable, could be subject to biases. Details on the selection of participants, the interface used for comparison, and the instructions provided to the participants should be included to ensure the study's robustness and reproducibility.

**Suitability:**

3

---

### Meta-Review · Senior_Area_Chairs · 2024-07-10

**Recommendation:** Accept (Poster)
**Confidence:** 4

**Metareview:**

This paper considers a method for text-guided image editing, specifically targeting small objects in the image which are often difficult to generate. Next to a new methodology a new benchmark dataset will be released. All reviewers agree that this is a paper that fits ACM Multimedia well. The reviews have gone through an interesting transition where one reviewer went from weak accept to borderline accept, where a borderline reject went to borderline accept. All in all we now have three borderline accepts. So, the paper can be accepted.